# Experimental and Numerical Case Studies about Two-Dimensional CHS Joints with a Symmetrical Y-Shape

**DOI:** 10.3390/ma15093179

**Published:** 2022-04-28

**Authors:** Patrick Heinemann, Dorina-Nicolina Isopescu

**Affiliations:** Department of Civil and Industrial Engineering, Faculty of Civil Engineering and Building Services, “Gheorghe Asachi” Technical University of Iasi, 1, Prof. Dr. Docent Dimitrie Mangeron Blvd. No. 59A, 700050 Iasi, Romania

**Keywords:** hollow sections, numerical simulation, welding line, construction, steel

## Abstract

Steel hollow section joints are mainly used for offshore structures, hall support or trusses. Current standards define different kinds of steel joints, but there are limitations regarding the geometry or load scenarios. Excluded joints are full-overlapped Y-joints with on-top connection. For these kinds of joints, there is no general design fundamental, so the design engineer has to build his/her own model. The aim of this paper is to figure out the resistance of this special undefined joint type and the influence of the inclination angle between the pipes. An experimental and numerical analysis of this joint was done. Due to this evaluation of the inclination angle, the design engineer could optimize the structure economically. Two different circular profile types were focused on. It was concluded that by increasing the inclination angle, the total applicable force decreased non-linearly. On one hand, the most economic design was achieved by choosing a steep angle. On the other hand, the resistance of the structure, regarding the maximum appliable force, could be increased. For the system in this paper, an increase up to 47% was achieved by choosing a profile with a thickness of 2.8 mm instead of 2.0 mm.

## 1. Introduction

Steel hollow section joints are mainly used for offshore structures, hall support or trusses. In the case of vertical structures, the joints can be formed by multiple variations, commonly plate-to-pipe connections or cast steel nodes. Under special conditions, there is the need for three-dimensional welded joints. Current standards, such as the Eurocode 3 [1] and Design Guides, such as the CIDECT [2], define welded joints. Some joint types, which are commonly used in the application field, which are named above, are excluded in these standards or literature. The full-overlapped Y-joint with on-top connection is not defined in these standards. The design engineer has to build his/her own model because there is no general design fundamental. This joint is characterized by a vertical pipe, which is defined as a chord. Two or more pipes are attached to the end of the chord. These pipes are defined as braces or branches. In contrast to cast steel, the advantage of this type of column is that it can be built directly on the construction side. These columns are either uniplanar or multiplanar. The pipes are cut inclined and connected by a three-dimensional welding line. To avoid an intermediate plate, equal cross–sections are chosen for the pipes. Due to the uniaxial arrangement, the resistance of the column decreases. The node, which is defined as the welded area of the pipes, is the weakest point at the joint. In the literature, much research focused on Y-shape steel joints. The difference between the fully overlapped joint in this paper and the standard-defined Y-joint mainly concerns the arrangement of the braces. In the standard, one or two branches are connected to the side flange of the chord. In the case of the Y-joint in this paper, the two braces are connected on top of the chord and there is an overlap between the chord and the braces. The design engineer has to build his/her own calculation model for this special kind of joint. At the same time, the aim of the design engineer is to archive a highly economical design of a structure. By variation of the geometry, for example, the inclination angle or steel profiles, the total applied force and the resistance can be influenced. Due to the numerical model in this paper, verified by experimental tests, the design engineer can economically optimize the structure of a uniplanar Y-shaped column with overlapped on-top connection.

It is difficult to find appropriate literature about this special joint type. This emphasizes the novelty and the relevance of the topic. The stress analyses are mainly conducted on variations of standard defined joints.

Numerical analyses of fully overlapped Y-joints are found in [3] for multiplanar joints and [4,5,6,7] for uniplanar joints. The branches are connected to the top of the chord. An intermediate plate is necessary because of the unequal diameters of the different pipes. This intermediate plate generates a different stress distribution than a directly connected system. The intermediate plate weakens the node [7]. If there is a steeper inclination angle, the contact area will become more concentrated. This results in higher stresses and deformations. Ibrahim et al. [8] conducted experimental studies regarding circular hollow sections under bending. Different geometrical parameters, such as the diameter-to-thickness ratio, were analyzed. It was concluded that wave buckling arises depended on the thickness of the profile. Jankovic et al. [9] conducted case studies about standard defined Y-joints with circular hollow sections. The inclined brace was connected to the side flank of a circular hollow section chord via a welding line. The effect of the brace was calculated, and the resultant stress was affected by the chord without modelling the complete joint. This was done for the compression force-load scenario. Mia et al. [10] conducted numerical studies about XT-joints combined with a fixed 90° inclination angle. This is a multiplanar joint with different load scenarios and profile thicknesses. The mesh is comparable to the numerical analysis presented in Section 4. It was concluded that the maximum stresses decreased with a higher profile thickness. Oshogbunu et al. [11] conducted numerical studies about multiplanar DKK-joints for offshore structures. The inclination angle was set to 45°, and the braces did not overlap. Different thickness ratios were investigated. For the numerical simulation, beam-solid 20-node elements were used. The focus was given to fatigue loadings. Ozyurt et al. [12] conducted case studies about different standard-defined circular-hollow section joints. T-, K-, X-, Y-, and N-joints were analyzed under the influence of tension and compression by elevated temperatures. There was a gap between the braces. The profile diameters and thickness varied. A refined mesh was set to the area of the welding line. For the compression case it was concluded that the strength calculation equations at ambient temperature delivered the most precise results. Zhao et al. [13,14] conducted experimental studies about overlapped K-joints. Different welding situations were considered. One brace was in tension, the other one was in compression. It was concluded that a hidden weld is recommended for overlapped joints. The throat thickness of the fillet weld and the weld quality in tubular joints should be strictly guaranteed in order to prevent cracking. Shen et al. [15] conducted experimental studies about T-,Y-, and K-joints with oval hollow section profiles. It was pointed out the importance of hollow section joints in the industry. It was concluded that the failure modes were mainly affected by the ratio of the brace diameter to the chord width and the intersection orientation. Punching into the chord increased failure.

However, this paper is about uniplanar steel joints with equal circular hollow section profiles (CHS) and a fully overlapped top-connection. The analysis was split into experimental and numerical parts to verify the results. The aim was to determine the influence of the inclination angle on the resistance of the joint. The influence of the profile thickness on the ultimate force was focused on. The ultimate force was characterized by rapidly decreasing resistance and an increasing force at the same time. To verify the quality of the welding line, non-destructive tests were executed. The specimens were prepared manually, not by robotics.

## 2. Materials and Methods

For the laboratory test, two different circular hollow section profiles were used. The sections differed in their thickness. The first profile had a circular hollow section (“RO”) 48.0 × 2.0 mm, which is defined in the standard EN 10219-2 [16]. The profile is named “CHS t = 2.0 mm” in the following chapters. The second cross–section was the RO 48.0 × 2.8 mm, which is defined in [16]. It is named “CHS t = 2.8 mm” in the following chapters. The notation “t” represents the thickness in the case of both profiles.

Every specimen was made of three pipes with a constant pipe length of 200 mm. The cross section was superimposable for all pipes. The pipes were cut in a three-dimensional way and welded with half of the original inclination angle. Due to this, no intermediate plate was necessary. For the analysis, four different inclination angles were prepared. The inclination angles were defined as 20°, 25°, 30°, and 45°. The welding line was a 3-mm-fillet weld, which represents the braces-to-brace and the braces-to-chord connection. Zamzami [17] showed the different kinds of welding lines that are appropriate for steel joints, and the advantages of the welding line types were highlighted. Due to the geometrical methods, friction welding or cruciform welding cannot be executed. The advantage of the fillet weld is the feasible implementation in a construction site. In view of the compression force machine, it is necessary to incline the free ends of the braces. Due to this, a bending component was induced next to the axial compression. Every test was repeated twice for verification. Overall, 4 (inclination angle) × 3 (verification) × 2 (profile types) = 24 tests were prepared. The material was the steel S 235, which was equal for all pipes. This steel type is defined by the following material properties [18] for thicknesses smaller 40 mm. The density was 7850 kg/m^3^, and Young’s modulus was 2.1 × 10^5^ MPa. The yield strength was 235 MPa, and the ultimate strength was 360 MPa. The Poisson’s ratio was 0.3. The experimental and numerical (Section 4) analyses were done in the elastic and plastic states. In the literature [19,20], numerical comparisons between different material types were executed. It was found that the common construction steel alloys did not have large differences in resistance compared to other material types, such as aluminum.

The load scenario was set as the compression force to imitate the load scenario resulting by a roof of a structure. For this, the specimen was positioned in a compression force machine. The testing machine was a Zwick/Roell SP1000 [21] axial compression/tension machine. The limit of this machine is 1000 kN. The dimensions of the specimens were adopted to the testing machine. Especially for the 45° model, the diameter of the bottom part of the machine must be large enough (Figure 1). The machine can be force or strain controlled. To test the ultimate load of the specimens, the machine was force-controlled for the experimental analysis. To avoid dynamic responses, the force speed was set to a low speed of 10 kN/min. To estimate the dynamic response, one test was executed with a speed of 1 kN/min. No dynamic response occurred for the 10 kN/min speed. Generally, cracks create a multiaxial and complex stress level in minimal areas, which are widely described in the literature [22,23,24,25]. Much research was done to calculate the crack levels. However, the tests were iterated to the plastic limit of the steel joint before a crack arose. Strain gauges were glued to the pipes at the positions, where the maximum stresses were expected to measure the stress distribution; 8 mm strain gauges with a single grid and an electric resistance of 120 Ohms were used. Because of the uniplanar system, a one direction measurement was set. The strain gauges could not be fixed directly on top of the welding line due to the round shape of the welding. However, the sensors were installed as close to the welding line as possible. The distance was 1–2 mm. The geometry of the specimen and the load scenario were symmetrical. For the case of small imperfections, a Teflon [26] plate was installed. By compressing the specimen into the elastic Teflon material, the imperfections were neglected. Because of this symmetry, one sensor was applied to the side of the chord member. Two strain gauges were installed in between both branches. Because of the symmetry and the Teflon plate, both sensors should record nearly superimposable results. Figure 1 shows the general setup for the laboratory test as a model in the case of the 45° system. Figure 2 shows the onsite situation.

Next to the strain gauges, a sensor was connected to the specimen to measure the horizontal deflects. This sensor is named “Linear Variable Differential Transformer” (LVDT). The linear and axial movement was converted by electric coils into an electric signal. The boundary conditions were set as simple supported. To block the global horizontal deflections, a Teflon [26] plate with a thickness of 10 mm was located under the specimen. When the compression force affected the joint, the ends of the braces punched to the plate without perforating the material. The horizontal sliding was blocked. Next to the axial compression, bending was induced due to the beveled ends of the branches.

Before the compression tests, a set of non-destructive tests was performed in the laboratory. The reason was to check the quality of welding lines because they were constructed by hand and not by robotics. Under large defects, the specimen will not be usable. In the first step, the specimens were visually inspected to detect imperfections and/or large defects regarding the welding line. In the second step, a liquid test was executed. Due to capillary forces, open cracks or pores were detected. The welding line was cleaned, degreased with the liquid U87, and taped. The penetrating liquid U88 was sprayed onto the surface. The penetration time was 25 min. The developer liquid U89 was applied. If any marks (open cracks in the surface) arose, the specimen had to be renewed. Third, to find deflects inside the welding line, the weld was inspected by ultrasonic waves (Figure 3). An ultrasonic pulser-receiver with 4 MHz and 10 mm diameter was used. This method is based on the ultrasonic reflection method. In Figure 3, the results of a measurement are shown as an example on the oscillator. After starting the test, calibration and finding the position, the amplitude decreased rapidly. If the amplitude was high relative to the calibration amplitude, deflects would be present and the specimen had to be renewed. Table 1 shows the devices used for the experimental tests. After the non-destructive tests were done, the compression tests were executed.

## 3. Results

### 3.1. Results of the Ultimate Limit Force for “CHS t = 2.0 mm”

No crack in the welding line occurred in the 24 specimens after performing the test. There was greater deformation in the area of the welding line. The inner position of the brace and chord moved together, while the outer area was pressed out. The global deformation is visualized in Figure 4.

In Figure 5, the results for the maximum compression force of the “CHS t = 2.0 mm” models are presented. The results are shown for the three tested samples and for each inclination angle system. The range of the differences of every test was small. The horizontal lines in the colors of each inclination model represent the average values. The average values were similar for the 25° and 30° models, including a delta of 9 kN. The differences between the 20°/25° and 30°/45° models were 16 kN and 18 kN. The standard deviation was as follows: 3.36 (20°), 5.05 (25°), 3.98 (30°) and 1.18 (45°). Figure 5 shows that the maximum compression force will become smaller if the inclination angle increases.

Figure 6 shows the stress–strain distribution of the circular hollow section profiles with 2.0 mm thickness, (CHS t = 2.0 mm). In combination with the cross–section area, the stress regarding the overall model was calculated. In Section 4, there is a comparison of the local stresses between the numerical and experimental model. In Figure 6, there are the results for every specimen (two results were lost). The average value for the three specimens was shown as a thicker graph, depending on the inclination angle. The range of the maximum strains was small. The different graphs, which represent the models, had a smooth behavior. The maximum resistance will decrease if the inclination angle increases. The behavior of the graphs for the stress–strain is similar to the findings from another experimental study [27].

### 3.2. Results of the Ultimate Force, CHS t = 2.8 mm

This paragraph is about the experimental analyzation of a fully overlapped Y-joint with on-top connection. The analyzation process and test execution are superimposable with Section 3.1. The profile thickness of the circular hollow section profile was enlarged from 2.0 mm to 2.8 mm. The overall deformations were smaller than those in the tests in Section 3.1 in the case of the larger thickness. Figure 7 shows the results for the compression forces for each tested specimen regarding the 2.8 mm CHS model depending on the four inclination angles. Because of the spreading in laboratory tests, each test was repeated twice. The horizontal lines in Figure 7 show the average values depending on the inclination angle. The differences of the average values for the 25° and 30° models had a small range including a delta of 6 kN. The differences of the 20°/25° and 30°/45° models were 29 kN and 34 kN. The standard deviation was as follows: 21.34 (20°), 19.04 (25°), 9.67 (30°), and 13.01 (45°). Similar to cases of the smaller thickness, the resistance will decrease if the inclination angle increases.

Figure 8 shows the stress–strain distribution in the case of the thicker CHS profile. In Figure 8, there are the results for every specimen. The average value for the three specimens was shown by a thicker graph depending on the inclination angle. The results of the 25° and 30° model were similar. There was a larger gap for the 20° and 45° results. Overall, the resistance of the steel joint will decrease if the inclination angle increases. 

### 3.3. Comparison of CHS t = 2.0 mm and CHS t = 2.8 mm

Figure 9 shows a comparison of the mean values of both CHS profile types. If the inclination angle increases, the total resistance of both joint types will decrease. The behavior of both graphs is nearly superimposable. Decreasing trends occurred for both graphs. The standard deviation was as follows: 24.04 (20°), 11.50 (25°), 14.62 (30°), and 6.48 (45°). However, there is a strong influence of the inclination angle on the resistance of the joint. The 2.8-mm-thick profile joint had significantly higher resistance than the 2.0-mm-thick profile joint. The gap between the graphs was about 47%.

## 4. Discussion

Numerical simulations were executed to find a feasible model for a design engineer to calculate fully overlapped steel joints with on-top connection. This study adopted geometrical models and material properties. The inclination angles were set to 20°, 25°, 30°, and 45°. The member lengths were superimposable. The numerical results were verified by the results of the experimental tests. The software Ansys [28] was used to simulate the Y-joints. The material was superimposable with the material in Section 3. The geometry was created by the use of three beveled pipes, connected by a 3D, 3-mm-fillet weld. The pipes had a gap of 1–2 mm including a phase. This ensured that there is no contact between the pipes. The force was only transferred through the welding line. For the numerical model, the contact between the pipes was set as frictionless to adopt this behavior. Due to the welding process, distortion in the microstructure can arise. This effect can result in an attenuation of the node. The heat-affected zone had a greater influence on the fatigue design of the node. The frictionless connection of the members was a numerical approach, which is on the safe side designing a node. Due to this, the effect of welding distortion was neglected in the numerical model. The boundary conditions were set as simple supported, combined with horizontal blocking of the braces’ ends. This imitated the effect of the Teflon [20] plate. The chord’s end was affected by an axial compression force, which was defined by the maximum compression of the laboratory tests. The value was calculated as an average result of three specimens. The mesh was set as 10 mm tetrahedral elements with a quadratic shape function. A refinement of 1.3 mm solid elements was set to the welding line area. In the literature, this mesh was verified for comparable systems [3,4,29]. To compare the quality of the mesh, a mesh metric evaluation was done. The geometrical conditions of the element shapes were compared to common limit values of the literature. In this case, the results of the “CHS t = 2.8 mm” with inclination of 25° model are presented. The skewness was less than 1 and tended to zero. The mean value was 0.273. The aspect ratio had a maximal value of 344, which is smaller than the common limit of 1000. The mean value was 2.04. The Jacobian value should be smaller than 30. The maximal value of the numeric model was 5.1, with an average of 1.05. Overall, the quality of the elements was 0.805. Due to the refinement of the mesh, the total number of elements was reduced. As an example, the number of nodes was 140,389 and the element number was 83,169 in the case of the 2.8 mm-30° model. Figure 10 shows the distribution of the von-Mises stresses on the surface of a specimen. The maximum stresses arose at the welding line between both braces. The deformation is shown in Figure 10. The deformation shape in the node and stress distribution was superimposable with the laboratory test results. Due to the force fitting of the specimen into the elastic 10 mm Teflon material, the measurements of the global deformation values were not comparable to the numerical results. The offset varied between the tests. The deformation shape of the node (Figure 4) fit the numerical result. The deformation was not centrally compressed, such as in experimental studies regarding standard defined T-joints [30]. Due to the round shape of the welding line, it was not possible to fix the strain gauge directly at the welding line. The strain gauges were fixed closely to the welding line. The stress/strain results were compared for this position. The values were measured for multiple positions, which are explained in Figure 1. Figure 11 shows the measuring data of the strain gauges 2 and 3 as an example for the different inclination angle models. The scatter was visualized for each specimen. The difference between the strain gauges was small due to the symmetrical geometry. Due to this, an average value was calculated for each system to compare it to the numerical model. Figure 12 shows a comparison between the results of the numerical and experimental analyses in the case of a position where the maximal stresses were expected. The results for the 25° modes were nearly superimposable. However, the numerical results were in good agreement with the experimental results.

Heinemann et al. [7] conducted studies about fully overlapped joints with on-top connection including an intermediate plate. The results regarding the maximal affected force are in contrast to the results in this paper. The stress distribution varied due to the different shape of the joints. The lines of application strike together in one point. The force was transferred through the welding line in the cross section of the chord. In the case of the model in [7], the intermediate plate took up the force and transferred it to the chord. The inclination angle had a different effect regarding the two models because of this different behavior of the joint.

Ibrahim et al. [8] conducted experimental studies regarding circular hollow sections under bending. The deformation shape differed from the results in this paper because of the different geometry. No wave buckling occurred in the experimental tests, though bending was part of the load case.

Jankovic et al. [9] conducted case studies about standard defined Y-joints by calculating the single members of the joint. This method is not applicable to a fully overlapped joint with on-top connection. The stress distribution, which arises due to the overlapping node, is complex. The braces generate force components, which are directly transferred to the cross section of the chord.

Mia et al. [10] conducted numerical studies about XT-joints under the effect of compression forces. The conclusion of the influence of the profile thickness on the stress distribution is similar to the results of this paper. Even though the geometrical shape was different for both models, the profile thickness affected the stresses significantly. In the case of the compression force scenario, the stress distribution in the area of the connection between brace and chord is similar to the results in this paper. The welding line was neglected in [10].

Models with equal member length were analyzed in this paper. In [7], the reduction in the resistance in the case of expanded member length was explained. Future studies will focus on the influence of unequal member length in combination with the inclination angle.

Oshogbunu et al. [11], Ozyurt et al. [12], and Shen et al. [14] conducted studies about standard defined steel joints under special conditions. The geometry and load cases were not comparable to the system described in this thesis. In the case of standard defined joints, the branches are in compression, while the chord is not affected by a load. This is not comparable to the full-overlapped Y-joint in this paper. All three members were in compression. A different stress state and distribution arose due to the different load case, geometry, and boundary conditions.

Zhao et al. [13] conducted experimental studies about overlapped K-joints. Both braces overlapped, but the connection to the chord was at the side flank of the chord. In combination with the opposite orientation of the axial loads, the deformation shape and stress distribution were not comparable to the special Y-joint in this paper. The conclusion concerning the welding quality was superimposable. Due to non-destructive pre-tests, the quality of the welding line was ensured.

As presented in this paper, the inclination angle had a large impact on the resistance of the special Y-joint made of steel. This is in contrast to other systems described in literature, but only under unique and unusual conditions. Azari-Dohan [19] found that under high temperatures (fire-induced loads) the inclination angle between the braces had no considerable influence on the static load for standard KT-joints.

## 5. Conclusions

After the experimental and numerical analyses of CHS joints including different profile thicknesses and inclination angles, the following conclusions were drawn:The total resistance decreased significantly if the inclination angle of a fully overlapped Y-joint increased. This conclusion is valid for the special geometrical parameters given in this paper. This behavior was analyzed for the inclination angles of 20°, 25°, 30°, and 45°.The decreasing behavior of the total resistance was non-linear. There was a greater effect for the extreme angles of 20° and 45°. The differences in results between the 25° and 30° system were small, but there was a decreasing trend. In future studies, more inclination angles in the range between 20° and 45° will be analyzed. To achieve an economically design, the engineer should choose a steep inclination angle. If this is not possible, because of architectural reasons or construction constraints, the resistance of the Y-shaped column will decrease non-linearly.The behavior of the 2.0-mm and 2.8-mm-thickness profiles was superimposable. The deformation shape and stress distributions were independent of the profile thickness if the force increased for the thicker profile.The CHS joint with a 2.8 mm thickness had higher resistant than that with 2.0 mm thickness. It is observed that, by increasing the thickness of 0.8 mm, the resistance increased by 47% on average. In future studies, different profile thicknesses will be analyzed to support this conclusion.The quality of the welding line was ensured by non-destructive pre-tests. Deflects and cracks outside and inside of the welding were eliminated. According to the literature, the deflects can have a large influence on the resistance of the node.There was good agreement between the numerical and the experimental results. The stress distributions were similar to the experimental results at relevant positions. The local strains and the shape of global deformations were superimposable for the numerical and experimental model. A feasible numerical model was presented, which helps engineers in practice to design a fully overlapped Y-joint. This special type of steel joint is undefined in the current standards including the geometrical and loading properties. The numerical model including the mesh choice was verified by experimental tests.

This analysis is part of a PhD thesis. In addition to the circular profiles, the behavior of the square, hollow section profiles with a similar cross-sectional area was focused on. It was assumed that, due to the edges of the squared profile, a different stress distribution was generated, and increasing joint resistance was highlighted.

## Figures and Tables

**Figure 1 materials-15-03179-f001:**
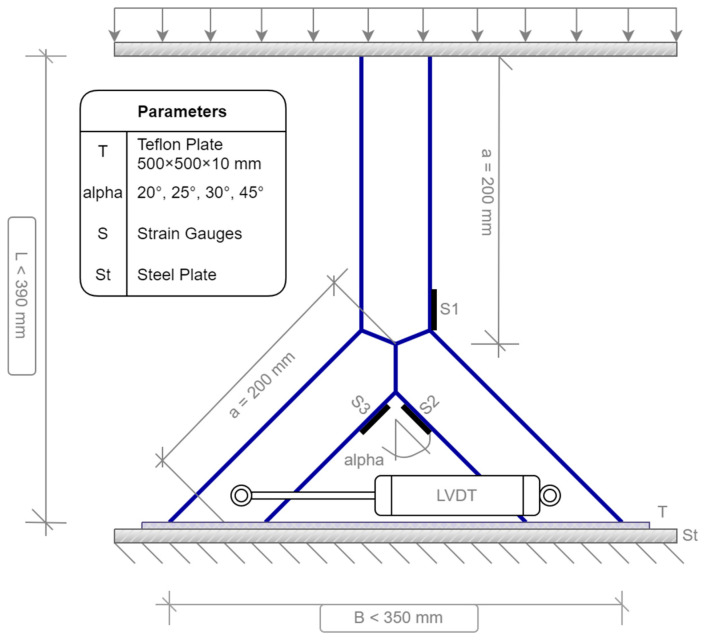
Laboratory test setup—model.

**Figure 2 materials-15-03179-f002:**
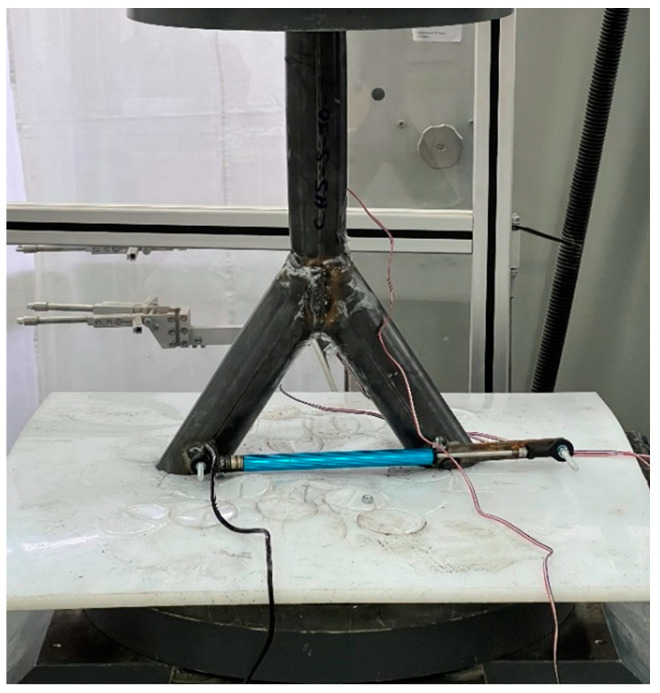
Laboratory test setup—onsite.

**Figure 3 materials-15-03179-f003:**
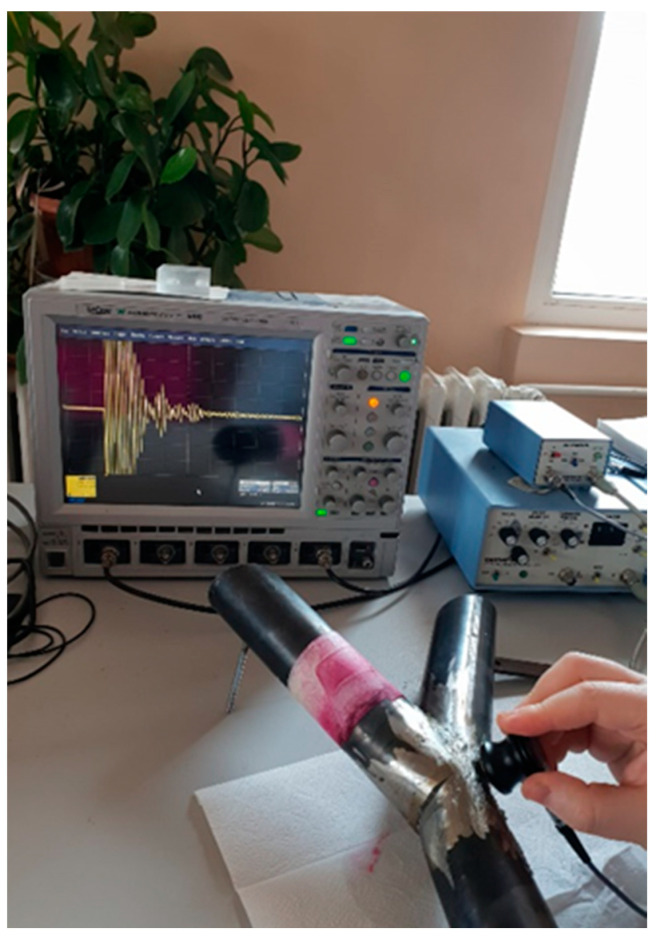
Non-destructive tests for welding lines.

**Figure 4 materials-15-03179-f004:**
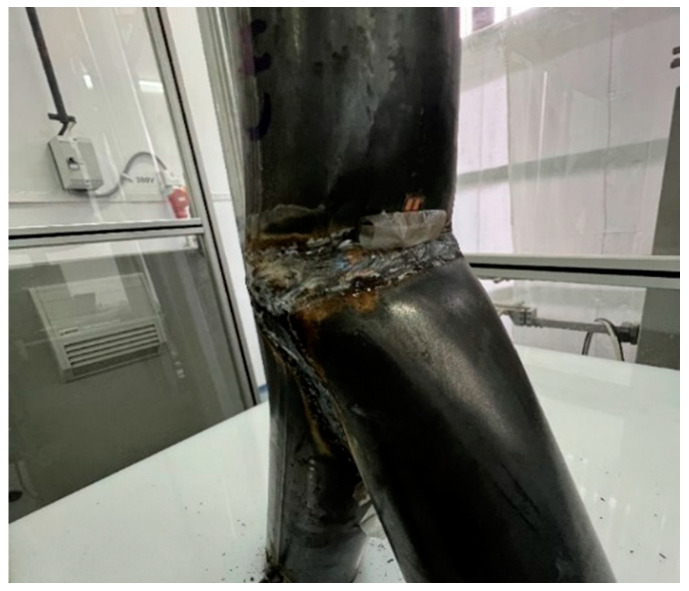
Deformation shape of the “CHS t = 2.0 mm” joint.

**Figure 5 materials-15-03179-f005:**
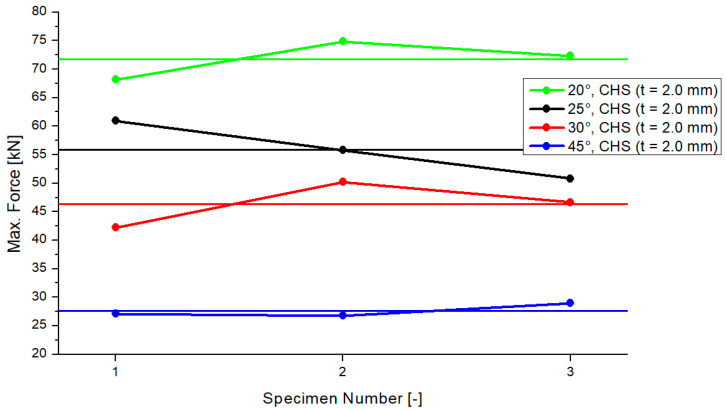
Results of maximum compression force, CHS t = 2.0 mm.

**Figure 6 materials-15-03179-f006:**
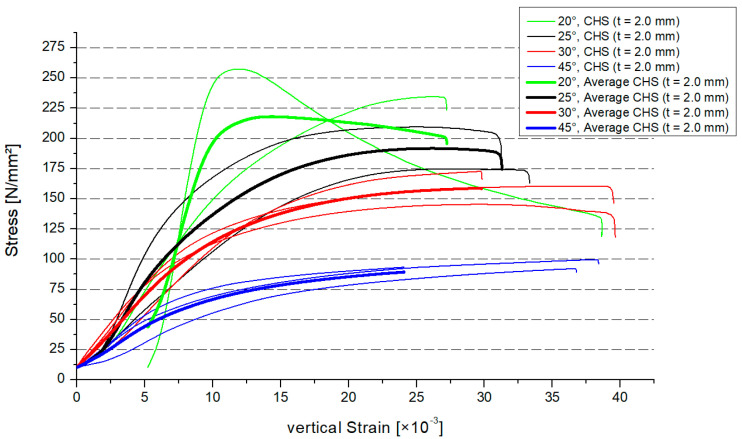
Stress–strain distribution, CHS t = 2.0 mm.

**Figure 7 materials-15-03179-f007:**
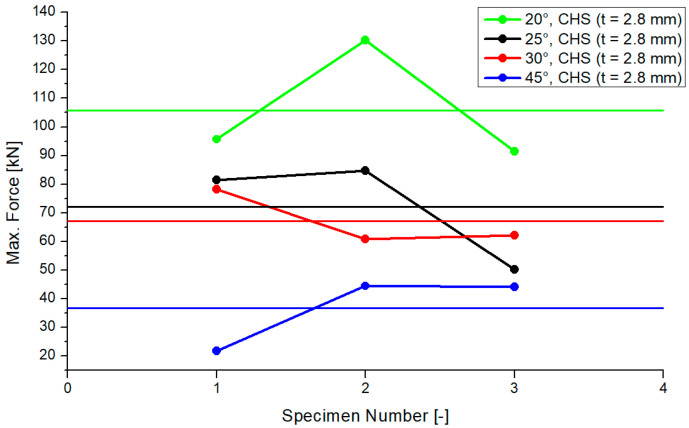
Results of the maximum compression force, CHS t = 2.8 mm.

**Figure 8 materials-15-03179-f008:**
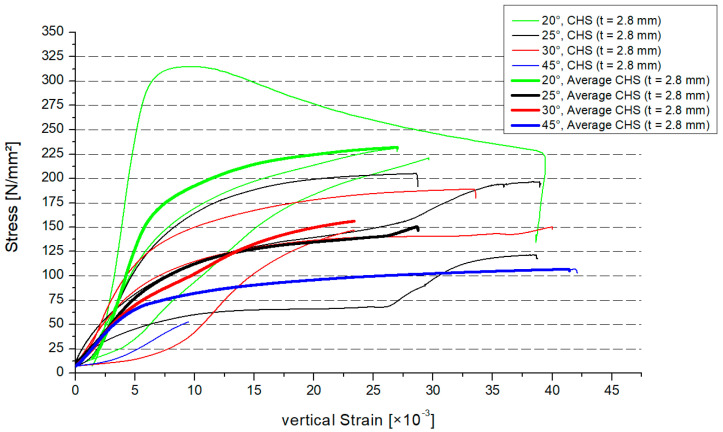
Stress–strain distribution, CHS t = 2.8 mm.

**Figure 9 materials-15-03179-f009:**
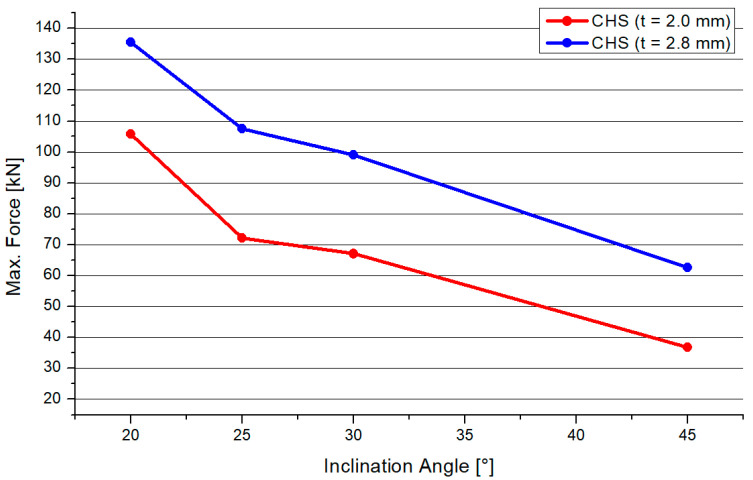
Comparison of the results of both CHS profiles.

**Figure 10 materials-15-03179-f010:**
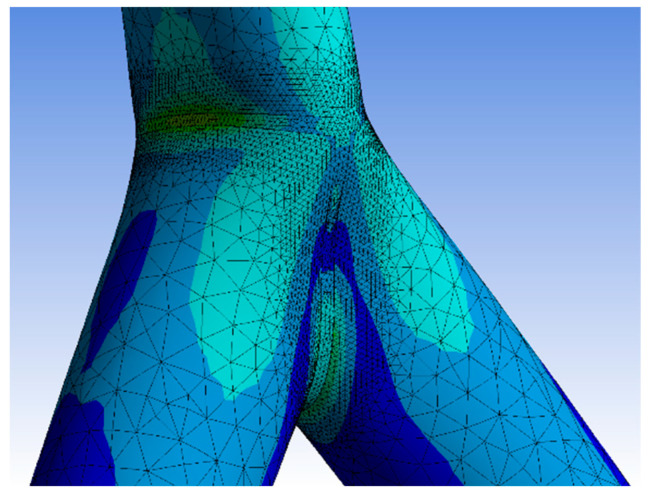
Von-Mises stress distribution and deformation.

**Figure 11 materials-15-03179-f011:**
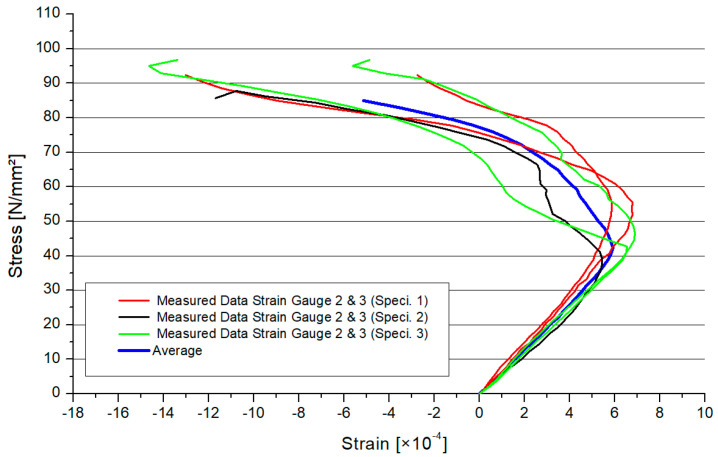
Von-Mises stress distribution and deformation.

**Figure 12 materials-15-03179-f012:**
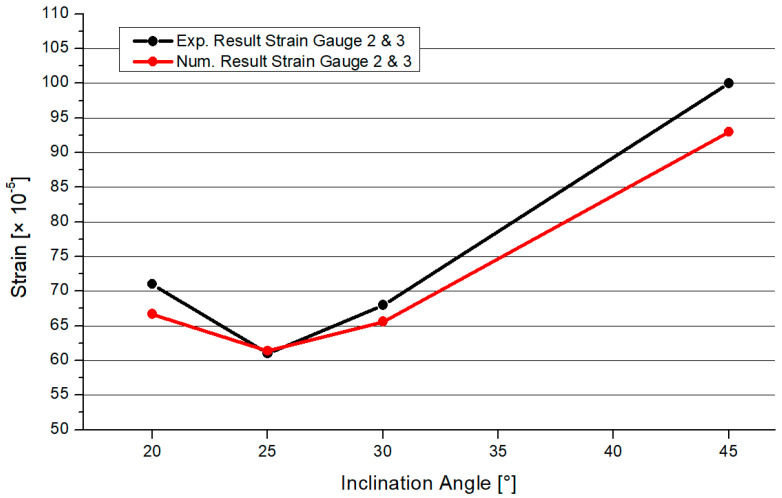
Comparison of the results of both CHS profiles.

**Table 1 materials-15-03179-t001:** Properties of the devices used in the experimental tests.

Name	Description
Zwick/Roell SP1000 [16]	Compression force testing machine
8 mm, 120 Ohms single grid	Strain Gauges
10 mm, 500 × 500 mm Teflon	Teflon Plate
Linear LVDT	Deflection Sensor
U87, U88, U89	Liquid Test
4 MHz, 10 mm Pulser-Receiver	Ultrasonic Test

## Data Availability

Not applicable.

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
