# Peer review of "Experimental and Numerical Case Studies about Two-Dimensional CHS Joints with a Symmetrical Y-Shape"

_materials, 2022, doi:10.3390/ma15093179_

Round 1

Reviewer 1 Report

line 74: What is LVDT? in Fig. 1?

line 74: Why are strain gauges not applied to both sides of the profiles? This makes it easier to detect asymmetrical behavior.

line 106: What type of strain gauge was used? Was measurement in one direction? Where was the exact position? How many millimeters from the weld?

line 150: What is the standard deviation for each angle?

line 152: There should be three curves for each angle in the diagram. Please still plot them. How was the total stress determined? The axis designation uses a comma instead of a point for the numbers.

line 173: The axis designation uses a comma instead of a point for the numbers.

line 182: The representation of the standard deviation for each angle is important. Please add standard deviation.

line 189: What material law did you use? Please specify the material parameters used.  Distortion is existent after welding. How was the geometric deviation due to welding taken into account in the model? 

line 223: Both strain gauges 2 and 3 behave the same. I do not believe that. Three specimens were tested. Where is the scatter in the experimental data?

What does the global deformation figure look like after the experiments and simulations? Local information (strains) was compared, but global (deformation) was not. Why were the values from the LVDT from the experiment not compared with the values from the simulation? 

Reviewer 2 Report

Dear Authors,

Congratulations on your work, which is focused on a very interesting subject. As any other paper in this phase, there are some amendments to do, whose can improve the overall quality of your paper. Thus, I'm providing below some comments and suggestions, trying to collaborate by this way in improving your paper:

  1. The Abstract doesn't clearly state the literature gap found, as well as the main motivation to develop this work. Thus, please clearly state the gap found in the literature in the Abstract, Introduction and Conclusions. The mains goals are also not clear in the Abstract.
    The novrlty brought by your work is also not properly pointed out. Thus, please state clearly the novelty that your paper represents for the scientific community, stating as well if your contribution is exclusively scientific or if there was some practical motivation behind the development of your work. Any industrial application based on this work should also be pointed out.
  2. Your Introduction is very good, mainly the second part (after reference [8]. However, the number of references is very scarce. Please try to deepen the study about previous works carried out in similar subsject.
  3. Figure 1 should be passed from the Introduction to the Materials & Methods section.
  4. Please correct the sentence: "Every specimen is made of three pipes...".
  5. Please point out the compression machine used, to let the reader realize the control exercized in the specimen.
  6. Please refer the main features of the devices (straing gauges) used in the deformation measurements.
  7. Please also characterize the microscope and the non-destructive techniques (equipment) used to control the weldings. Please refer the rejection criteria.
  8. Please insert a table with all conditions used in the tests performed.
  9. In Materials & Methods, you state that: "After verifications, it was concluded, based on the requirements stipulated in norms, that the arise small imperfections do not influence the total resistance of welding lines and/or of the joints, significantly.". Please show evidences of this through macro images of the small defects.
  10. In Figure 4, the welding bead seems non-uniform. I would like to see the welding closer, because this is fundamental to ensure that the linkage is in good state.
  11. Please include information about the type of elements used in simulation, the software used, and the number of elements/nodes considered.

Hope these comments help you in improving your paper.

Kind regards.
